# Dietary practice and associated factors among elderly people in Northwest Ethiopia, 2022: Community based mixed design

**Mulat Tirfie Bayih** *, **Adane Ambaye Kassa, Yeshalem Mulugeta Demilew**

School of Public Health, Department of Nutrition and Dietetics, College of Medicine and Health Science, Bahir Dar University, Bahir Dar, Ethiopia

* mulatbonny@gmail.com

## Abstract

### Background

The planet's population is aging at an incredible speed. Poor dietary practices are a major problem among the elderly. However, literature is scarce on dietary practices among elders in the study area. Therefore, the results of this study may give information to decision-makers.

### Objective

This study aimed to assess dietary practices and associated factors among elderly people in Northwest Ethiopia, in 2022.

### Methods

A community-based mixed study design was employed among elderly people from May 20 to July 2, 2022. Systematic random sampling and purposive sampling techniques were used for selecting study participants in quantitative and qualitative studies, respectively. Data were collected using an interviewer-administered structured questionnaire for the quantitative part and an interview guide for the qualitative part. Binary logistic regression analysis was used. A P-value less than or equal to 0.05 was used to declare statistically significant variables. A crude odds ratio and an adjusted odds ratio with a 95% confidence interval were used to measure the strength of the association. Thematic analysis was used for qualitative data analysis.

### Results

A total of 422 participants were recruited for the study. Twenty-six in-depth interviews were done. The prevalence of adequate dietary practice was only 54.5% [95% CI: (49.8, 59.2)]. It was significantly associated with being aged between 65 to 74 years (AOR: 8.32; 95 CI: 3.9, 18.1), being aged between 75 to 84 years (AOR: 2.90; 95% CI: 1.1, 7.9), eating sometimes alone (AOR: 1.86; 95% CI: 1.03, 3.4), eating always with family members (AOR: 4.96; 95% CI: 2.6, 9.4), and food security (AOR: 3.13; 95% CI: 1.8, 5.4). Thematic analysis revealed

**Data Availability Statement:** All relevant data are within the manuscript.

**Funding:** The author(s) received no specific funding for this work.

**Competing interests:** The authors have declared that no competing interests exist.

**Abbreviations:** AOR, Adjusted Odds Ratio; CI, Confidence Interval; COR, Crude Odds Ratio; FANTA, Food and Nutrition Technical Assistance; FCS, Food Consumption Score; HFIAS, Household Food Insecurity Access Scale; HFSS, Household Food Security Status; NCD, Non-Communicable Diseases; PCA, Principal Component Analysis; SPSS, Statistical Package for Social Science; WFP, World Food Program; WHO, World Health Organization.

three themes that interfere with the dietary practices of elders. A majority of in-depth interviewees mentioned that there were taboos and cultural beliefs which favor inadequate dietary practices of the elderly; the study participants reported that individual, economic, societal, and physiological factors are barriers affecting the dietary practices of the elderly, and all respondents have no experiences regarding elderly dietary practices.

## Conclusion

The prevalence of adequate dietary practice was low. It was significantly associated with age, with whom feeding, and household food security status. Taboos and cultural beliefs, barriers, and experiences hampered the dietary practices of elders. Therefore, improving the dietary practices of elders focusing on advanced age, loneliness, food security, taboos, cultural beliefs, barriers, and experiences regarding dietary practices should be done.

## Introduction

The planet Earth's population is aging at an incredible speed. The number and percentage of older adults are rising in all countries, and these trends are expected to continue [1, 2]. The proportion of adults aged 65 and older in the global population (9.3%) would have surpassed that of children under the age of five (8.7%) in 2020. By 2050, the proportion of people aged 65 and older (15.9%) will be more than double that of children under the age of five (7.1%), and will also outnumber young adults aged 15 to 24(13.7%) [2]. The life expectancy at birth in Ethiopia showed an increment from 51 year of age in 2000 to 65 years of age in 2021 [3]. Factors such as community based health strategies, improving access to safe water, female education and gender engagement, and the rise of civil society organization play a crucial role in these life expectancy increment [4]. According to the Ethiopian population census commission report of 2007, the proportion of population aged 65 years and above was 3.2% [5].

As more people reach old age, they are more vulnerable to poor dietary practices due to changed sensory functions (eg. taste and smell), poor oral health and functionality, change of satiety, difficulty in preparing foods, limited financial resources, loss of role functions loneliness and social isolation [6]. Older people are more prone than younger people to suffer from a variety of age-related diseases and functional impairments that can make it difficult to maintain a healthy diet practice [7]. Poor dietary practice is a problem in the elderly people and it is major risk factor for under nutrition and for Non-communicable diseases, psychological problems, low cognitive functions and unhealthy aging [8–12]. Various researchers published findings regarding elders' dietary practices. Poor dietary practice was estimated to be 53.3% in Ghana and in Nepal 100% of old age consume vegetables whereas 17.8% consumed meat and only 1% consumed an egg [8, 13].

The Ethiopian government endorsed the national food and nutrition policy and strategies to solve the nutritional problems including the older ages by accommodates lifecycle approach and dietary practices have gotten attention [14, 15]. HelpAge international and Mekedonia humanitarian association attempted to improve elderly dietary practice and health by food provision services, health care service, hygiene facilities, shelter, clothes provision services, recreational facilities and funeral ceremonies when they died and [16, 17].

In the global food system, food and nutrition security continues to be a significant concern. In developing countries, limited resources, drought, political instability, less advanced

agriculture and infrastructure challenges with many people lacking consistent access to enough nutritious food. As a result diet variety and quality becomes less and low, often relying on staple grains like rice tefe and maize. On the other side, developed countries typically have efficient and stable food systems, with advanced agricultural technologies, solid supply chains, and comprehensive food safety standards. This leads in a broad and high-quality food supply, with a large variety of fruits, vegetables, meats, and grains available all year round [18–20].

Dietary practice is defined as the observable action or behavior of dietary habits. They are the quantities, proportions, variety, or combinations of different foods and beverages in diets and the frequency with which they are habitually consumed. It is a complex interplay of physiologic, psychological, social, and genetic factors that influence meal timing, quantity of food intake, food preference, and food selection. They can also be referred to as the habitual decisions of individuals or groups of people regarding what foods they eat [21, 22]. The frequency of meal and food consumption score (FCS) was used to assess the dietary practices of the elderly [8, 21].

Even though the elderly population is considerable, Ethiopia's policies, plans, programs, and projects have not prioritized their nutrition., Rice, teff, lentils, sorghum, maize, oats, chickpeas, nuts, other cereals, other legumes, fruits, and vegetables are the common food sources in the area. Even though Fish and other seaweeds are found in the area, they are not the usual foods in the community due to literacy level and culture [23]. There is limited information in Ethiopia in general and particularly in the study area about the dietary practice and associated factors among elderly people. Thus, by presenting empirical evidence, this research will persuade the governmental and nongovernmental communities to integrate older issues into their policies, strategies, programs, and projects. The present study is carried out to identify factors associated with inadequate dietary practice in elderly people, fill information gaps, and suggest possible problem-solving approaches. It is also essential to identify the level of dietary practice and factors among elderly people since the results may give information to develop and inter into nutrition intervention activities for Woreta town administration health office to optimize dietary practice for the elderly people.

## Methods and materials

### Study setting and period

The study was conducted at Woreta town administration from May 20 to July 2, 2022. Woreta, the town of Fogera district, is 55 and 625 kilometers away from Bahir Dar and Addis Ababa respectively. Woreta is located east of Lake Tana, west of Farta district, north of Dera district, and south of Libo Kemkem district. The weather condition is categorized as Woyna Dega [24].

Woreta town has 4 kebeles, the smallest administrative units, with a total population of 55,043. Woreta town administration has 1933 elderly people. It has one health center and four health posts [25].

### Study design and populations

A community-based cross-sectional study was used for the quantitative part and phenomenology was used for qualitative part among elderly people. All elderly people age greater than or equal to 65 years in the town administration were included for both quantitative and qualitative part.

### Sample size determination

The required sample size was computed based on a single population proportion formula assuming the prevalence (p) of dietary practice 50% (because there was no previous study

conducted in Ethiopia), 95% confidence level (1.96), and 5% margin of error.

$$n = (Z\alpha/2)^2 \frac{p(1-p)}{d^2}$$

Where; n = estimated sample size, $Z\alpha/2$ = a standard normal value which corresponds 95% of confidence level = 1.96, p = estimated prevalence of dietary practice is 0.5, d = tolerable sampling error = 0.05. Finally by taking 10% of non-response rate, the sample size is 422.

For qualitative part, in-depth semi-structured interviews were conducted among 26 elderly individuals. The interviews ceased when data saturation was achieved since in determining the sample size of the qualitative research, what matters was theoretical data saturation [26].

## Sampling techniques and procedures

A systematic random sampling technique was used to select the study participants. The total sample size was proportionally allocated for all kebeles based on their catchment number of elderly people to obtain the required number of study participants.

The sampling interval of households in each kebele was determined by dividing the total number of households which has eligible elderly people to the allocated sample size meaning to select the study units (elderly people). Systematic random sampling was used with sampling interval $k^{th}$ which was calculated as k = 1933/422 = 4.58 = 5.

Out of the first 5 elderly people, the initial one was determine by using the lottery method. For households with multiple elderly people, one elderly person was selected by lottery method.

Based on this, every five participant was interviewed until the required sample size was fulfilled. For in-depth interview, participants were chosen using a purposive sampling technique.

## Operational definitions

**Adequate dietary practice.**   When the elderly has at least three meals daily and acceptable food consumption score (FCS); whereas inadequate dietary practice: when the elderly has less than three meals daily or/and unacceptable FCS [8, 21, 27, 28].

**Acceptable FCS.**   When the elderly's FCS is >35; whereas unacceptable FCS: when the elderly's FCS is 0–35 [29, 30].

**Social support.**   The help provided by family, friends, groups, or communities. A person who is considered as social support gives a support of any aid, healthcare service, and support on food preparation and washing clothes [31].

**Elderly.**   People whose age is $\geq$ 65 years [32].

## Data collection procedures and variable measurement

An interviewer-administered, pre-tested, and structured questionnaire was used to collect the quantitative data. The questionnaire was constructed into eight major parts, such as socio-demographic and economic related factors, psychosocial related factors, activities of daily living related factors, health related characteristics, meal related factors, food consumption score (FCS) related characteristics, wealth index related characteristics, behavior related factors, and household food security status (HFSS) related factors.

The Food Consumption Score is a composite score based on dietary diversity, food frequency, and the relative nutritional importance of different food groups. The interviewee was asked about the frequency of consumption (in days) over a recall period of the past 7 days. Food items are grouped into eight standard food groups with a maximum value of seven days

per week. The consumption frequency of each food group is multiplied by an assigned weight that is based on its nutrient content. Those values are then summed to have FCS [33].

Food consumption score is defined as poor food consumption score: 0 to 21, borderline food consumption score: 21.5–35, and acceptable food consumption score: > 35. To assess FCS, the participants were asked to recall the foods they consumed in the previous seven days before the survey. Each food item is given a score of 0 to 7, depending on the number of days it is consumed. Food items were grouped into food groups, and the frequencies of all the food items surveyed in each food group were summed. Any summed food group frequency value over 7 was recorded as 7. For each participant, the food consumption score was calculated by multiplying each food group frequency by each food group weight and summing these scores into one composite score [34].

The household wealth index was computed using indicators for semi-urban residents. The wealth status was determined through principal component analysis (PCA). The responses of all non-dummy variables were classified into two parts. Codes were given as 1 for the highest scores and 0 for the lower scores. In PCA, those variables having a communality value greater than 0.5 were used to produce factor scores. Finally, the score for each household on the first principal component was used to create the wealth score. Terciles of the wealth score were created to categorize households as poor, medium, and rich.

The household food security status (HFSS) of participants was measured using the Food and Nutrition Technical Assistance (FANTA) household food insecurity access scale (HFIAS). It is specifically designed to measure households' limited access to food in the previous month. The tool is constructed from nine food insecurity occurrence questions with 27 frequencies of occurrence. A HFIAS score variable is calculated for each household by summing the codes for each frequency of occurrence question. The maximum score for a household is 27 (the household responded to all nine frequency of occurrence questions with "often", coded with a response code of 3); the minimum score is 0 (the household responded "no" to all occurrence questions; frequency of occurrence questions are skipped by the interviewer and subsequently coded as 0 by the data analyst). The higher the score, the more food insecurity the household experienced. The lower the score, the less food insecurity a household experiences. Then, HFSS was categorized as "food insecure" (when a household scored 2) and "food secure" (when a household scored 1) [34].

The qualitative data were collected using a semi-structured guide. It was performed using an in-depth interview with participants recruited from the community. The data were collected by the principal investigator.

## Data quality control

To maintain the quality of data, first, standardized data collection tools were developed in English and translated to Amharic (local language) for data collection then back to English for consistency. Training was given for the supervisor and data collectors for one day on data collection tools, ethics, and approaches to interviewing techniques. Regular supervision and reviewing the completed questionnaire on a daily basis were done by the investigator and the supervisor to maintain the quality of the data. Pretest was done on 5% of the total sample size in other sites in order to evaluate the developed questioner. During data collection, questionnaires were reviewed and checked for completeness by the supervisor and principal investigators and the necessary feedback was offered to the data collectors in the next morning. During qualitative data collection, tape recordings and field notes were taken. Transcription was done on the same day of data collection. To assure trustworthiness, the qualitative data were

triangulated with the quantitative data. We also used the purposive sampling technique to enhance its trustworthiness.

## Data processing and analysis

Data were cleaned and entered into the Epi-Data version 3.1 statistical software and were exported to Statistical Package for Social Science (SPSS) statistical software package, version 26 for analysis. Before analysis, missing values and outliers were checked. Frequency and cross tabulations were used to summarize descriptive statistics. Frequencies, proportions, means, and standard deviations were used to summarize variables; and presented using texts, tables, and figures. A binary logistic regression model was fitted to identify factors associated with dietary practice. Variables with a p-value of < 0.20 in the bi-variable analysis were fitted into the multivariable logistic regression analysis. Both crude odds ratio (COR) and adjusted odds ratio (AOR) with the corresponding 95% confidence interval (CI) was calculated to measure the strength of association. Finally, a p-value of < 0.05 was used to determine statistically association. Model fitness was checked using Hosmer Lemishow test (p = 0.113).

Atlas.ti7 software was used to quote, code, to make theme (family), and analysis qualitative data. Thematic analysis was used for qualitative data analysis.

## Ethical consideration

Ethical clearance was obtained from ethical Review Board of Bahir Dar University College of Medicine and Health Sciences (protocol number 475/2022). Permission was obtained from Woreta town and each kebele administration. All methods were carried out in accordance with relevant guidelines and regulations. Before collecting the data, written informed consent was obtained from each study participants. Each study participants were informed about the purpose of the study and participation was voluntary without payment for their participation. Each study participants also were informed that the right to withdraw at any time during the interview. All gathered information were protected from its confidentiality, anonymity was explaining clearly to participant. Except the principal investigator information is not exposed third person. Interviewing and recording was not done without permission from interview participants.

## Results

### Socio-demographic, economic, and psychosocial related characteristics of the elderly people

A total of 422 people aged ≥ 65 years participated in this study with a response rate of 100%. Sixty-four percent of respondents were females. The mean (±SD) age of the respondents was 74.26(±8.67) years. More than half (54.0%) of the respondents were married. About sixty-five percent (65.2%) of the respondents were Orthodox Christians in their religion. Seventy-eight percent of respondents were illiterate. Regarding credit services, about two-thirds (66.4%) of the respondents had no access to credit services. Nearly one fourth (23.9%) of respondents didn't have social support. (Table 1).

### Health and Behavioral related characteristics of the elderly people

About 32.5% and 7.3% of the respondents had a history of chronic diseases and acute illness during the interview respectively. Regarding chronic disease types, 56(13.3%) had hypertension, 47(11.1%) diabetic mellitus, 6(1.4%) tuberculosis, and 7(1.7%) HIV/AIDS. Almost thirty-five percent (35.3%) of the respondents took drugs for treatment for acute and chronic diseases

**Table 1. Socio-demographic, economic, and psychosocial related characteristics of study participants in Woreta town, Northwest Ethiopia, 2022.**

| Variables | | Frequency (n = 422) | Percent (%) |
|---|---|---|---|
| Sex | Male | 152 | 36.0 |
| | Female | 270 | 64.0 |
| Age in years | 65–74 | 262 | 62.1 |
| | 75–84 | 94 | 22.3 |
| | ≥85 | 66 | 15.6 |
| | Mean (±SD*) | 74.26(±8.67) | |
| Family size | family size < 4 | 186 | 44.1 |
| | family size ≥ 4 | 236 | 55.9 |
| Marital status | Married | 228 | 54.0 |
| | Widowed | 171 | 40.5 |
| | Divorced | 23 | 5.5 |
| Religion | Orthodox | 275 | 65.2 |
| | Muslim | 135 | 32.0 |
| | Protestant | 12 | 2.8 |
| Living with whom | With Partner | 239 | 56.6 |
| | With Children | 141 | 33.4 |
| | Alone | 25 | 5.9 |
| | Others* | 17 | 4.0 |
| Educational status | Unable Read And Write | 221 | 52.4 |
| | Read And Write | 108 | 25.6 |
| | Primary School (1–8) | 39 | 9.2 |
| | Secondary School (9–12) | 31 | 7.3 |
| | Diploma And Above | 23 | 5.5 |
| Occupation | Housewife | 221 | 52.4 |
| | Merchant | 77 | 18.2 |
| | Employed | 28 | 6.6 |
| | Daily Laborer | 21 | 5.0 |
| | Farmer | 8 | 1.9 |
| | Retired | 67 | 15.9 |
| Head of the household | Father | 251 | 59.5 |
| | Mother | 110 | 26.1 |
| | Children | 44 | 10.4 |
| | Relative | 17 | 4.0 |
| Access to credit services | Yes | 142 | 33.6 |
| | No | 280 | 66.4 |
| Pension user | Yes | 67 | 15.9 |
| | No | 355 | 84.1 |
| Access to CBHI* | Yes | 258 | 61.1 |
| | No | 164 | 38.9 |
| Social support | Yes | 321 | 76.1 |
| | No | 101 | 23.9 |
| Wealth index | Poor | 137 | 33.2 |
| | Medium | 143 | 33.4 |
| | Rich | 142 | 33.4 |
| Food secured | Yes | 283 | 67.1 |
| | No | 139 | 32.9 |

SD* = Standard deviation; others* = relative; CBHI* = Community based health insurance

**Table 2. Activities of daily living related factors of study participants in Woreta town, Northwest Ethiopia, 2022.**

| Variables | | Frequency (n = 422) | Percent (%) |
|---|---|---|---|
| Able to walk outside | Yes | 362 | 85.8 |
| | No | 60 | 14.2 |
| Able to bath alone | Yes | 361 | 85.5 |
| | No | 61 | 14.5 |
| Able to carry things | Yes | 319 | 75.6 |
| | No | 103 | 24.4 |
| Able to shop | Yes | 322 | 76.3 |
| | No | 100 | 23.7 |
| Able to pick up things | Yes | 330 | 78.2 |
| | No | 92 | 21.8 |
| Able to stand up alone | Yes | 374 | 88.6 |
| | No | 48 | 11.4 |

during the interview. About 2.6%, 17.8%, and 15.6% of the respondents had a history of ciga-
rette smoking, alcohol use, and chew khat during the interview respectively.

## Activities of daily living related characteristics of the elderly people

More than three-quarters of the older people were able to undertake activities of daily living,
such as being able to stand up by themselves (88.6%), walk outside (85.8%), take a bath by
themselves (85.5%), pick up objects (78.2%), and carry objects (75.6%). Table 2.

## Dietary practices and meal related characteristics of the elderly people

Nearly half of the respondents, 54.5% (95% CI: 49.8, 59.2) had adequate dietary practice.
Nearly fifty-five percent (54.7%) of respondents took their meal regularly (breakfast, lunch,
and dinner). Table 3. The majority of the respondents consumed Staples (97.9%), oils (87.7%),
pulses (74.6%) and vegetables (70.1%) seven times per week. Fig 1. Majority of the respondents
(85.1%) had acceptable food consumption score. Fig 2.

## Factors associated with dietary practices of the elderly people

In the bi-variable logistic regression analysis, sex, age, access to credit services, able to walk
outside, able to bath alone, able to carry things, able to pick up things, able to stand up alone,
with whom feeding, and food security status had a p-value of < 0.20. In the multivariable logis-
tic regression analysis, age, with whom feeding, and food security status were remained signifi-
cantly associated with dietary practices.

The odds of adequate dietary practice were 8.32 times (AOR: 8.32; 95 CI: 3.9, 18.1) higher
among elders aged 65 to 74 years compared to 85 years and above. As well, the odds of

**Table 3. Dietary practice of the elderly people in Woreta town, Northwest Ethiopia, 2022.**

| Variables | | Frequency (n = 422) | Percent (%) |
|---|---|---|---|
| Dietary practice | Adequate dietary practice | 230 | 54.5 |
| | Inadequate dietary practice | 192 | 45.5 |
| Frequency of meal/day | Less than three meals per day | 191 | 45.3 |
| | Three or more meals per day | 231 | 54.7 |

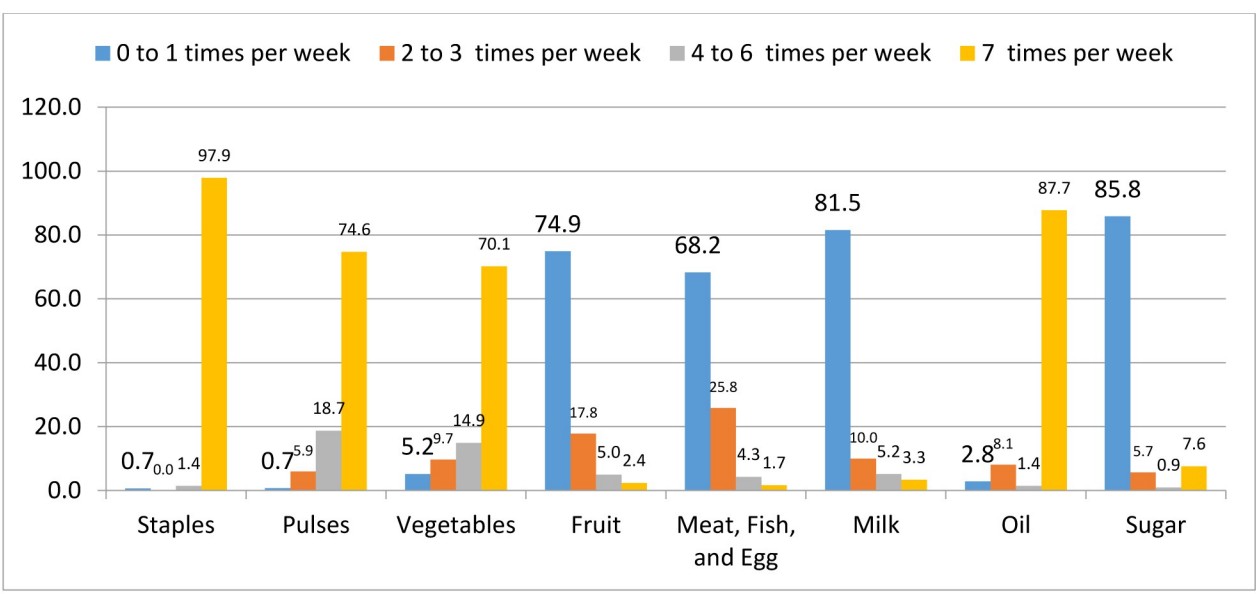

**Fig 1. Frequency of consumption of various food groups by percentage in Woreta town, Northwest Ethiopia, 2022.**

adequate dietary practice were nearly 3 times (AOR: 2.90; 95% CI: 1.1, 7.9) higher among elders aged 75 to 84 years compared to 85 years and above. Likewise, the odds of adequate dietary practice were 1.86 times higher among people eating sometimes alone (AOR: 1.86; 95% CI: 1.03, 3.4) than eating always alone. Also, the odds of adequate dietary practice were nearly

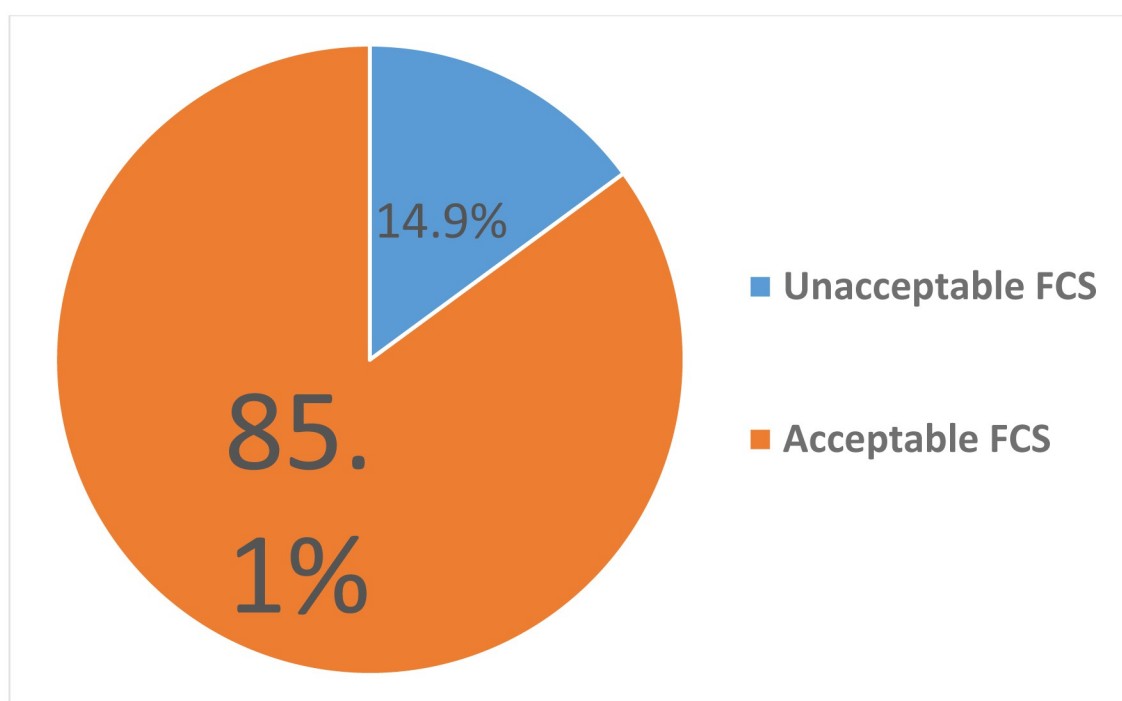

**Fig 2. Food consumption score the elderly people in Woreta town, Northwest Ethiopia, 2022.**

**Table 4. Factors associated with dietary practices of the elderly in Woreta town, Northwest Ethiopia, 2022.**

| Variables | | Dietary practice | | COR (95% CI) | AOR (95% CI) |
|---|---|---|---|---|---|
| | | Adequate No (%) | Inadequate No (%) | | |
| Sex | Male | 96(63.2) | 56(36.8) | 1.74 (1.2,2.6) | 1.16(0.7,1.9) |
| | Female | 134(49.6) | 136(50.4) | 1 | 1 |
| Age in years | 65–74 | 178(67.9) | 84(32.1) | 10.60(5.3,21.3 | 8.32(3.9,18.1)* |
| | 75–84 | 41(43.6) | 53(56.4) | 3.87(1.8,8.3) | 2.90(1.1,7.9) |
| | ≥85 | 11(16.7) | 55(83.3) | 1 | 1 |
| Access to credit services | Yes | 99(69.7) | 43(30.3) | 2.62(1.7,4.0) | 1.23(0.7, 2.1) |
| | No | 131(46.8) | 149(53.2) | 1 | 1 |
| Able to walk outside | Yes | 216(59.7) | 146(40.3) | 4.86(2.6,9.2) | 0.95(0.2,3.9) |
| | No | 14(23.3) | 46(76.7) | 1 | 1 |
| Able to bath alone | Yes | 219(60.7) | 142(39.3) | 7.01(3.5,13.9) | 1.67(0.4,6.4) |
| | No | 11(18.0) | 50(82.0) | 1 | 1 |
| Able to carry things | Yes | 203(63.6) | 116(36.4) | 4.93(3.0,8.1) | 4.22(0.9,19.9) |
| | No | 27(26.2) | 76(73.8) | 1 | 1 |
| Able to pick up things | Yes | 206(62.4) | 124(37.6) | 4.71(2.8,7.9) | 1.54(0.4,6.0) |
| | No | 24(26.1) | 68(73.9) | 1 | 1 |
| Able to stand up alone | Yes | 218(58.3) | 156(41.7) | 4.19(2.1,8.3) | 1.13(.2,5.3) |
| | No | 12(25.0) | 36(75.0) | 1 | 1 |
| Feeding with | Always alone | 36(30.3) | 83(69.7) | 1 | 1 |
| | Sometimes alone | 96(56.8) | 73(43.2) | 3.03(1.8,5.0) | 1.86(1.03,3.4)* |
| | Always with family Members | 98(73.1) | 36(26.9) | 6.28(3.6,10.8) | 4.96(2.6,9.4)* |
| Food security status | Food secure | 183(64.7) | 100(35.3) | 3.58(2.3,5.5) | 3.13(1.8,5.4)* |
| | Food insecure | 47(33.8) | 92(66.2) | 1 | 1 |

COR = Crude odds ratio; CI = Confidence interval; AOR = Adjusted odds ratio;

* = P-value <0.05

five times higher among people eating always with family members (AOR: 4.96; 95% CI: 2.6, 9.4) than eating always alone. Furthermore, the odds of adequate dietary practice were 3.13 times higher among those food secure participants (AOR: 3.13; 95% CI: 1.8, 5.4) than food insecure participants. (Table 4).

## Qualitative findings

Three themes and five sub-themes were identified by the qualitative study findings. These themes are taboos and cultural beliefs regarding dietary practices, barriers of dietary practice and experiences with regard to elderly dietary practice of the elderly people.

**Theme 1: Taboos and cultural beliefs.** A majority of in-depth interviewees mentioned that there were taboos and cultural beliefs which favor inadequate dietary practices of the elderly. Taboos and cultural beliefs were major factors that affect dietary practices of the elderly.

*Perception*. Some of the elderly food preferences are different from the age lower than adults and less.

> *Elderly people do not grow so that they should not eat nutritious food. Nutritious food should be given to the growing population segment (a 74 years old man). If a young man eats, he will defend anything; and if he catches, he will be snatched away (a 66 years old nun).*

*Religious reasons.* Elderly people are more engaged in their supernatural power. In doing so there are activities that hamper adequate dietary practices.

*Eating too much is forbidden in Holly Kuran (a 65 years old Muslim man). I used to eat meat, poultry, and egg; since being monk I had stopped to eat them; it is enough after being monk (an 86 years old monk).*

Fasting prevented the elderly from eating animal products like milk and milk products, meat, egg, and fish. It also prevented them from having more meals per day. Most of the respondents had only two meals during the fasting periods.

*I eat meat, egg, and fish in non-fasting periods. Also I did not have three meals (a male 65 year old retired man).*

**Theme 2: Barriers.**  The study participants reported that individual, economic, societal, and physiological factors are barriers affecting the dietary practices of the elderly. They could be presented by sub-themes as:

*Economic factors.* All of the qualitative study participants stated that food insecurity was the main constraint of having adequate dietary practice.

*My pension is run out in 15 days; after the 15th day of my pension people may help me; other ways l may go to my bed without any food (A retired 74 years old man). I eat milk, egg, and meat in festive seasons only (75 years old woman). If there are foods, I eat three meals per day (a 65 years old nun).*

Almost all of the qualitative study participants explained that unavailability of money was a major factor that decreases adequate dietary practice of the elderly people.

*I eat banana if children bring it to me. I drink milk if my child brings it to me (a widowed 68 years old woman). I always eat shiro because there are no cattle which produce milk. If there are lentils, I need to eat lentils. I can't buy lentils. I eat meat only during festive season. I usu-ally eat only one meal especially during fasting periods (a married 82 years man).*

*Societal factors.* Almost all of the qualitative study participants stated that loneliness affected their dietary practice.

*My grandchild supports me. If she is not found in the house, I will not eat anything alone (a widowed 68 years old woman). I live alone. My food is prepared by house rentals. If they didn't prepare my food, I go to my bed without food. If I get somebody who prepares my food, I eat three meals. If not, I eat only one meal (a female 75 years nun).*

A qualitative study participant explained that being dependent negatively affected her die-tary practice.

*Since I live with children, I do not want to ask the food what I want (a widowed 70 years old woman).*

Majority of the qualitative study participants reported that no social support for the elderly was the main constraint of having adequate dietary practice.

*I can't get breakfast timely because there is no person that helps us to prepare food . . . . . . .. I always eat lunch and dinner (A retired 65 years man).*

Interpersonal relationship refers to the elderly residents' relationships established with their companions such as their families, friends or other residents. One respondent expressed that a food item which is disliked by his intimate friend also disliked by him.

*I do not eat lentil for long period of time because my intimate friend didn't like to eat it (an 86 year old monk)*

*Physiological factors*. Almost all of the qualitative study participants stated that physiological factors affect their life in lowered their physical capacity.

*I did not go to church which is around 100 meters away from my home (a widowed 75 years old woman). Most of my teeth are lost. I eat "injera fitfit" (an* 86 *monk). . ..*

Most of the qualitative study participants stated that loss of appetite was a factor which decreases adequate dietary practices of the elderly.

*I used to finish one" injera", currently I can't finish a half of it (a widowed 68 years old woman). Elderly need tasty foods because they are with loss of appetite. Even though elderlies need many meals as children, I have loss of appetite (an 85 years old nun).*

Majority of the qualitative study participants stated that illness were factors which hamper adequate dietary practices of the elderly.

*I can't eat all food items (soft drinks, salt, and meat) since I have hypertension and gastritis (a widowed 70 years female interviewee).*

**Theme 3: Experiences with regard to elderly dietary practices.** All study participants in the in-depth interviews mentioned that there were inadequate dietary practices of the elderly.
A 70 years old woman in-depth interviewee said: *I always eat "injera with shiro". The amount and frequency of meals are less during elderly compared with the people younger than me.. . .*

## Discussions

Understanding the dietary practice and associated factors among elderly people is crucial for developing appropriate nutritional interventions. In this study only 54.5% (95% CI: 49.8, 59.2) of respondents had adequate dietary practices. This was similarly reported by the qualitative study participants that support the existence of inadequate dietary practices in the elderly people. A seventy year old woman reported that;

*I always eat injera with shiro. The amount and frequency of meals are less during elderly compared with the people younger than me (a 70 years old woman).*

This is in disagreement with studies conducted in Ghana and Nepal [8, 27]. This might be due to the study area is a surplus production main staple food groups. A study done in Kiambu County, Kenya [9] revealed a lower result than this study report. The disagreement may be due to differences in the study area since the current study was conducted among urban residents who might have the opportunity of access to nutrition education. It might also be the study area is a surplus production area.

Nearly fifty percent, 45.3%, of the study participants consumed less than three meals per day. This study was much lower than a study done in Meru County, Kenya [35]. This might be due to most (65.2%) respondents in this study were Orthodox in their religion. My supplementary qualitative findings revealed Orthodox followers had inadequate dietary practices since they skip at least their breakfast during fasting seasons. Fasting prevented the elderly from eating animal products like milk and milk products, meat, egg, and fish. It also prevented them from having more meals per day. A 65 years old retired man reported that:

*"I eat meat, egg, and fish only in non-fasting periods. Also I did not have three meals during fasting periods (a 65 year old retired man).*

In addition, majority (97.9%) of the elderly took staples foods seven times per week. However, they occasionally took animal source foods which is consistent with a study in Sir Lanka consumption of meat is low [36]. This may be due to food insecurity status and access of variety food items. This was also supported by qualitative study findings. A 74 years old man and a 75 years old woman reported that:

*My pension is run out in 15 days; after the 15th day of my pension people may help me; other ways l may go to my bed without any food (A retired 74 years old man). I eat milk, egg, and meat in festive seasons only (75 years old woman). If there are foods, I eat three meals per day (a 65 years old nun).*

Minority of (3.3%) of the study participants had milk and milk products seven times per week. Only 1.7% of the study participants ate meat, egg, and fish seven times per week. These findings have a huge discrepancy with study findings done in Meru County, Kenya, Bangladesh, and Switzerland [35, 37, 38]. This might be due to lower socio-economic status-poor and medium wealth index and food insecurity was 66.6% and 32.9% respectively- of the study participants.

In this study, the likelihood of having adequate dietary practice was increased with decreasing elderlies' age. Qualitative study participants reported smaller appetites and requiring smaller portion sizes are common in advanced ages. This finding is in consonance with findings of studies conducted in Ghana, Meru County, Kenya, and Malaysia [8, 35, 39]. It might be due to change in physiological or psychological conditions like loss of taste, appetite and smell, change of dental status and swallowing difficulties which are similar in all globe's elder. Elderly people may also be more susceptible to the gradual loss of physical capacities including mobility and movement. This may result in changes in food choice, eating habits and dietary intake, subsequently raising the risk of inadequate dietary practice. Finally, elderly people make less healthy food choice as they get older [11].

This study also revealed that the likelihood of having adequate dietary practices were higher among elderly people who ate their meal with family members than ate alone. The qualitative findings supported that eating alone is the common factor for inadequate dietary practice in the elderly people.

*My grandchild supports me. . . . . . .If she is not found in the house, I will not eat anything alone (a widowed 68 years old woman).*

This result is consistent with the study done in Thailand [40]. However, it is in disagreement with some findings of a scoping review conducted by Björnwall A. et al, 2021 [41]. It may be due to the Ethiopian elderly people did not want to live and eat alone because they have experienced many social life in their early ages and show lack of interest in food and eating when they eat alone. Also, if they eat with other people, they are willing to cook and eat all meals for the sake of others. In addition, the problem gets amplified when living alone and eating alone are combined since people who live alone did not get a chance to eat with others. Ethiopian elders give special value for having meals with others and it is a special event for them.

In this study, food security was the other significantly associated factor for adequate dietary practice. This finding is supported by the qualitative study findings. Almost all of the qualitative study participants reported that the most major factor for inadequate dietary practice was food insecurity.

*My pension is run out in 15 days; after the 15th day of my pension people may help me; other ways l may go to my bed without any food (A retired 74 years old man).*

This finding is in consonance with a review done by Nicola Veronese and Stefania Maggi [11]. Likewise, it is similar with a study done in Thailand [40]. This could be due to that food costs are increased across the globe in which elders can't afford the current food prices. Then, this might ultimately lead to inadequate dietary practice among the elders.

There was no significant association between sex and dietary practices of elderly people in this study. The reason for this might be equal support and attention given for both genders in this age group. In this study, access to credit services was not also a predictor for the dietary practices of elders. This might be due to the reason that access to credit service is not a primary concern for this age group. The other probable reason for this is elderlies may fear for back payment of the debt and as result they do have lower appetite to be credit service user.

## Strength and limitation

This study was done by using mixed study design method which enables method triangulation between quantitative and qualitative methods. It has also 100% response rate. When using this report, the study's limitation should be kept in mind. Even though we gave training for the supervisor and the data collectors on how to use probing and local events to minimize bias, there could be a chance to commit recall bias since feeding practice measurements rely on memory. The other potential limitations of this study include the possibility of self-reported data biases, the cross-sectional nature of the study (which may limit causal inferences), and the possibility of non-generalizability to other regions.

## Conclusions and recommendations

The prevalence of adequate dietary practice was low among the elderly people. Adequate dietary practice was significantly associated with age, with whom feeding, and household food security status. The qualitative study findings showed perception, religious reasons, economic factors, societal factors, and physiological factors were barriers that interfere with dietary practices of the elderly people. We recommend to the community and family members of the elderly to give nutritional and social support activities for the elderly people. It is also

recommended to researchers to do further research by using longitudinal methods which may minimize recall bias and give strong evidence.

## Acknowledgments

We would like to offer our in-depth gratitude to the data collectors, participants for their support to us.

## Author Contributions

**Conceptualization:** Mulat Tirfie Bayih, Adane Ambaye Kassa.

**Data curation:** Mulat Tirfie Bayih, Adane Ambaye Kassa, Yeshalem Mulugeta Demilew.

**Formal analysis:** Mulat Tirfie Bayih, Adane Ambaye Kassa.

**Investigation:** Adane Ambaye Kassa.

**Methodology:** Mulat Tirfie Bayih, Adane Ambaye Kassa.

**Software:** Adane Ambaye Kassa.

**Supervision:** Mulat Tirfie Bayih, Yeshalem Mulugeta Demilew.

**Validation:** Mulat Tirfie Bayih, Adane Ambaye Kassa.

**Visualization:** Adane Ambaye Kassa.

**Writing – original draft:** Mulat Tirfie Bayih.

**Writing – review & editing:** Mulat Tirfie Bayih, Yeshalem Mulugeta Demilew.

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
