## [Decision Letter · Decision Letter 0]

19 Jan 2024

PONE-D-23-36778Dietary practice and associated factors among elderly people in Northwest Ethiopia, 2022: Community based mixed design.PLOS ONE

Dear Dr. Bayih,

Thank you for submitting your manuscript to PLOS ONE. After careful consideration, we feel that it has merit but does not fully meet PLOS ONE’s publication criteria as it currently stands. Therefore, we invite you to submit a revised version of the manuscript that addresses the points raised during the review process.

Specifically, the reviewer recommends expanding upon the background regarding dietary advice and lifespans in Ethiopia. They recommend providing additional details, such as main food sources/choices, lifestyle, occupational and religious characteristics which could also influence diet. They also request clarification regarding the small sample size.

We look forward to receiving your revised manuscript.

Kind regards,

Jennifer Tucker, PhD

Staff Editor

PLOS ONE

Journal Requirements:

3. Please include your tables only as part of your main manuscript and remove the individual files. Please note that supplementary tables (should remain/ be uploaded) as separate "supporting information" files"

Additional Editor Comments:

Please note that we have only been able to secure a single reviewer to assess your manuscript. We are issuing a decision on your manuscript at this point to prevent further delays in the evaluation of your manuscript. Please be aware that the editor who handles your revised manuscript might find it necessary to invite additional reviewers to assess this work once the revised manuscript is submitted. However, we will aim to proceed on the basis of this single review if possible. 

Reviewers' comments:

Reviewer's Responses to Questions

**Comments to the Author**

1. Is the manuscript technically sound, and do the data support the conclusions?

Reviewer #1: Yes

2. Has the statistical analysis been performed appropriately and rigorously? 

Reviewer #1: Yes

3. Have the authors made all data underlying the findings in their manuscript fully available?

Reviewer #1: Yes

4. Is the manuscript presented in an intelligible fashion and written in standard English?

Reviewer #1: Yes

5. Review Comments to the Author

Reviewer #1: This study is really interesting because it is one of the first diet practical studies in this country.

Related with it, I have to say some recommendations to improve this paper:

1- In the background-introduction, I recommend to amplify the references number. You mentioned some importants aspects in diet policy.

2- I think that you must talk about lifespan in Ethiopia, and also the main differences between develop and undeveloped countried in the food frange.

3- You have to comment also main food and lifestyle characteristics of the study population, because only with food profile we could not understand well some differences between religion, occupation, etc.

4- Maybe, I suggest that they could explain better why they did qualitative part in only 26 person and what factors or characters are relevant in this 26 person. Maybe results could be different in they way of this qualitative test.

5- Also, I suggest to add an image about FCS.

6. PLOS authors have the option to publish the peer review history of their article (what does this mean?). If published, this will include your full peer review and any attached files.

Reviewer #1: No

---

## [Author Response · Author response to Decision Letter 0]

27 Jan 2024

We addressed all the comments and questions raised by the reviewer point by point.

---

## [Decision Letter · Decision Letter 1]

18 Jun 2024

PONE-D-23-36778R1Dietary practice and associated factors among elderly people in Northwest Ethiopia, 2022: Community based mixed design.PLOS ONE

Dear Dr. Bayih,

Thank you for submitting your manuscript to PLOS ONE. After careful consideration, we feel that it has merit but does not fully meet PLOS ONE’s publication criteria as it currently stands. Therefore, we invite you to submit a revised version of the manuscript that addresses the points raised during the review process. General comment: strictly adhere with the journals submission gudline. Give continuous line numbers to easily locate comments. For example, the fourth paragraph of the first sentence, written as "In the global food system," is an incomplete sentence or inappropriate punctuation.

Abstract: Include the 95% CI for the prevalence of adequate dietary practice.

Method: the sampling procedure is unclear. What was your primary sampling unit? households or individual elderly? What was the total number of households in the town? Of which, what was the total number of households with at least one elderly person? As you know, in Ethiopian culture, when the grandparents are old enough, they jointly live in their children's house ("tibatie") with a monthly or more round for each available child. How did you address them? Even, how the K value is applied is unclear? is it for the household or for each individual? If so, the K value is 4.58, which is rounded to 4, but you selected every two. How did you select, if both couples were elderly?

Result: Modify the interpretations; do not interpret OR as RR. For example, "Elders aged 65 to 74 years were 8.32 times more likely to have adequate dietary practice (AOR:

8.32; 95 CI: 3.9, 18.1) than elders aged greater than or equal to 85 years". It would be better if you interpreted the odds of adequate dietary practice as 8.32 times (AOR:

8.32; 95 CI: 3.9, 18.1) higher among elders aged 65 to 74 years compared to 85 years and above.

Table 4: Locate the reference categories with a value of 1 rather than making the cells empty.

We look forward to receiving your revised manuscript.

Kind regards,

Agegnehu Bante

Academic Editor

PLOS ONE

Journal Requirements:

Reviewers' comments:

Reviewer's Responses to Questions

**Comments to the Author**

1. If the authors have adequately addressed your comments raised in a previous round of review and you feel that this manuscript is now acceptable for publication, you may indicate that here to bypass the “Comments to the Author” section, enter your conflict of interest statement in the “Confidential to Editor” section, and submit your "Accept" recommendation.

Reviewer #2: All comments have been addressed

Reviewer #3: All comments have been addressed

2. Is the manuscript technically sound, and do the data support the conclusions?

Reviewer #2: Yes

Reviewer #3: Yes

3. Has the statistical analysis been performed appropriately and rigorously? 

Reviewer #2: Yes

Reviewer #3: Yes

4. Have the authors made all data underlying the findings in their manuscript fully available?

Reviewer #2: Yes

Reviewer #3: Yes

5. Is the manuscript presented in an intelligible fashion and written in standard English?

Reviewer #2: Yes

Reviewer #3: Yes

6. Review Comments to the Author

Reviewer #2: Dear Author,

Thank you for the nice work. I have minor comment for improvements

1. Why one admstrative town considered and why not didn't considered the rural area of the district. This need critical justification.

2. The result should be discussed with similar study conducted elsewhere and looks like shallow.

3. Make sure that no study conducted in Ethiopia and you took 50% during sample size calculation

Reviewer #3: General comment:

The article is well structured, with a clear rationale, robust methodology and comprehensive analysis. It makes a valuable contribution to understanding the dietary practices of older people in northwestern Ethiopia. However, future research could benefit from longitudinal designs, validation of data collection instruments, and exploration of additional regions for wider applicability.

Specific comments:

Methods:

- Data Collection: There is no information in the document as to whether the tools have been validated or whether there have been any pre-testing procedures.

Results:

-It is recommended that the clarity of tables and figures be enhanced and that qualitative findings are presented in more detail in the abstract and discussion.

Discussion:

- Potential limitations of this study include the possibility of self-reported data biases, the cross-sectional nature of the study (which may limit causal inferences), and the possibility of non-generalizability to other regions.

- It is recommended that the findings be integrated in the discussion in order to provide a more holistic view. This integration should include both qualitative and quantitative findings.

7. PLOS authors have the option to publish the peer review history of their article (what does this mean?). If published, this will include your full peer review and any attached files.

Reviewer #2: No

Reviewer #3: No

---

## [Author Response · Author response to Decision Letter 1]

19 Jun 2024

Comments and questions were addressed.

---

## [Decision Letter · Decision Letter 2]

7 Jul 2024

PONE-D-23-36778R2Dietary practice and associated factors among elderly people in Northwest Ethiopia, 2022: Community based mixed design.PLOS ONE

Dear Dr. Bayih,

Thank you for submitting your manuscript to PLOS ONE. After careful consideration, we feel that it has merit but does not fully meet PLOS ONE’s publication criteria as it currently stands. Therefore, we invite you to submit a revised version of the manuscript that addresses the points raised during the review process.

We look forward to receiving your revised manuscript.

Kind regards,

Agegnehu Bante

Academic Editor

PLOS ONE

Journal Requirements:

Additional Editor Comments:

You addressed all the comments raised in the previous round of revisions. Your manuscript needs one last proofread. In the previous revision, I suggested some modifications to the interpretation, and you did it well. However, it still needs some modifications. Eg., line 269, "The odds of adequate dietary practice as 8.32 times (AOR: 8.32; 95 CI: 3.9, 18.1) higher among elders aged 65 to 74 years compare 85 years and above.

Actually, it was my mistake too. So, please modify as: The odds of adequate dietary practice were 8.32 times (AOR: 8.32; 95 CI: 3.9, 18.1) higher among elders aged 65 to 74 years, compared to 85 years and above. Modify all the rest with the same or better approach.

Reviewers' comments:

Reviewer's Responses to Questions

**Comments to the Author**

1. If the authors have adequately addressed your comments raised in a previous round of review and you feel that this manuscript is now acceptable for publication, you may indicate that here to bypass the “Comments to the Author” section, enter your conflict of interest statement in the “Confidential to Editor” section, and submit your "Accept" recommendation.

Reviewer #2: All comments have been addressed

Reviewer #3: All comments have been addressed

2. Is the manuscript technically sound, and do the data support the conclusions?

Reviewer #2: Yes

Reviewer #3: Yes

3. Has the statistical analysis been performed appropriately and rigorously? 

Reviewer #2: Yes

Reviewer #3: Yes

4. Have the authors made all data underlying the findings in their manuscript fully available?

Reviewer #2: Yes

Reviewer #3: Yes

5. Is the manuscript presented in an intelligible fashion and written in standard English?

Reviewer #2: Yes

Reviewer #3: Yes

6. Review Comments to the Author

Reviewer #2: Dear Author,

Thank you for considering all my comments and I suggested for acceptance to publish your work. Please keep up the good work.

Reviewer #3: All errors and suggestions have been corrected and implemented by the authors. In my opinion, the answers are convincing and this article has no serious flaws and can be printed.

7. PLOS authors have the option to publish the peer review history of their article (what does this mean?). If published, this will include your full peer review and any attached files.

Reviewer #2: **Yes: **Habtamu Fekadu Gemede (PhD)

Reviewer #3: No

---

## [Editor Report · Decision Letter 3]

11 Jul 2024

Dietary practice and associated factors among elderly people in Northwest Ethiopia, 2022: Community based mixed design.

PONE-D-23-36778R3

Dear Mr. Bayih,

We’re pleased to inform you that your manuscript has been judged scientifically suitable for publication and will be formally accepted for publication once it meets all outstanding technical requirements.

Kind regards,

Agegnehu Bante

Academic Editor

PLOS ONE
---

## [Editor Report · Acceptance letter]

16 Jul 2024

PONE-D-23-36778R3 

PLOS ONE

Dear Dr. Bayih, 

I'm pleased to inform you that your manuscript has been deemed suitable for publication in PLOS ONE. Congratulations! Your manuscript is now being handed over to our production team.

Kind regards, 

on behalf of

Mr. Agegnehu Bante 

Academic Editor

PLOS ONE